# Three-Dimensional (3D) Visualization under Extremely Low Light Conditions Using Kalman Filter

**DOI:** 10.3390/s23177571

**Published:** 2023-08-31

**Authors:** Hyun-Woo Kim, Myungjin Cho, Min-Chul Lee

**Affiliations:** 1Department of Computer Science and Networks, Kyushu Institute of Technology, 680-4 Kawazu, Iizuka-shi, Fukuoka 820-8502, Japan; 2School of ICT, Robotics, and Mechanical Engineering, Hankyong National University, IITC, 327 Chungang-ro, Anseong 17579, Kyonggi-do, Republic of Korea

**Keywords:** digital image processing, integral imaging, Kalman filter, photon-counting integral imaging, volumetric computational reconstruction

## Abstract

In recent years, research on three-dimensional (3D) reconstruction under low illumination environment has been reported. Photon-counting integral imaging is one of the techniques for visualizing 3D images under low light conditions. However, conventional photon-counting integral imaging has the problem that results are random because Poisson random numbers are temporally and spatially independent. Therefore, in this paper, we apply a technique called Kalman filter to photon-counting integral imaging, which corrects data groups with errors, to improve the visual quality of results. The purpose of this paper is to reduce randomness and improve the accuracy of visualization for results by incorporating the Kalman filter into 3D reconstruction images under extremely low light conditions. Since the proposed method has better structure similarity (SSIM), peak signal-to-noise ratio (PSNR) and cross-correlation values than the conventional method, it can be said that the visualization of low illuminated images can be accurate. In addition, the proposed method is expected to accelerate the development of autonomous driving technology and security camera technology.

## 1. Introduction

Recently, research on three-dimensional (3D) imaging technology has been undertaken by many researchers [1,2,3,4,5,6,7,8,9,10,11,12,13,14,15,16,17,18,19,20,21]. In the field of research, there are interpretations of 3D information using technologies such as LiDAR, stereo vision, and so on. Moreover, 3D imaging technology has been used in various applications such as computed tomography (CT) scans and distance measurement sensors for autonomous driving technology. Representative 3D imaging technologies are stereo vision [1,7,8,9,10] and holography [5,11,13,14,15,16], although there are various techniques for interpreting 3D information. Stereo vision technology is a 3D imaging technology that uses binocular parallax to obtain the depth information of an object, and applied technology fields include augmented reality, virtual reality, autonomous driving and autostereoscopic display. Stereo vision has a simple system structure and it can obtain 3D information with high speed. Holography is a 3D imaging technology that uses the coherence and phase information of light that cannot be recognized by the human eye or image sensor. We can obtain depth information of an object using an interference pattern generated by overlapping two lights of the same wavelength. In addition, holography can record 3D information with full parallax and continuous viewing points from a single shot. However, they have several problems. Stereo vision has the problem that little 3D information may be obtained from the recorded images due to its binocular nature, and holography has the problem that special instruments such as a laser and a beam splitter are required. Therefore, a technique with accurate 3D information and a simple structure is required.

As a technique to solve these problems, integral imaging, which was first proposed by G. Lippmann in 1908 [22], is an analysis technique of 3D information reconstructed from multi-viewpoints images. It can provide full parallax and continuous viewing points without any special viewing devices and coherent light sources. Thus, it may be applied in various research fields such as 3D imaging [23], pattern recognition [24], image encryption and decryption [25], and so on. To obtain more accurate 3D information in these various research fields, the visual quality of elemental images is the most important factor. However, when this technology is applied under extremely low light conditions, it is difficult to obtain accurate 3D information. Extremely low light conditions can be defined as an environment in which it is difficult to distinguish objects with an image sensor or the human eye due to a small number of photons. To solve this problem, photon-counting integral imaging [26] has been proposed. It is 3D imaging method under photon-starved conditions that calculates the existence probability of a photon from the captured image by Poisson random process. The captured image under these conditions may not be recognized since it does not have sufficient amount of light. In the conventional photon-counting integral imaging [26], it had been performed on multi-perspective images captured in an extremely low light environment, and it had reconstructed 3D images by volumetric computational reconstruction (VCR) [26] to analyze 3D information and achieve high-resolution 3D images. However, the photon-counting integral imaging has a problem that reconstructed results are always random due to the temporal and spatially independent Poisson distribution. For this reason, it is difficult to obtain accurate 3D object information.

To solve this problem, we propose a method of applying Kalman filtering to photon-counting integral imaging. A Kalman filter can estimate a constant value from observations and gain values. That is, it predicts values that will occur in the future [27]. Therefore, it can be used to estimate distance and human motion in image processing research. In addition, the advantage of a Kalman filter is that the analysis includes prediction error covariance matrices that vary over time according to the statistics of the underlying model and the assimilated observations. For this reason, Kalman filtering has been applied many research fields [28,29,30,31]. In this paper, we regard multiple random reconstructed results of photon-counting integral imaging as time-series data with errors, and corrected the data with the Kalman filter. Therefore, it is expected that more accurate visualization results can be obtained.

This paper is organized as follows. In Section 2, we present computational reconstruction in integral imaging such as volumetric computational reconstruction, photon-counting integral imaging, and Kalman filtering. Then, image processing and the experimental setup are described in Section 3. To verify our proposed method, we show the experimental results with a discussion in Section 4. Finally, we conclude with a summary in Section 5.

## 2. Theory

### 2.1. Volumetric Computational Reconstruction (VCR)

Integral imaging is a technology that uses multi-perspective images to obtain 3D information. A concept of integral imaging is illustrated in Figure 1.

As illustrated in Figure 1, integral imaging can be divided into two processes: the pickup process and the reconstruction process. In the pickup process, multiple 2D images with different viewpoints (i.e., perspectives) recorded through imaging devices such as a lenslet array, pinhole array, or camera array for 3D objects can be obtained, where these images are referred to as elemental images. Three-dimensional information can be obtained by calculating the distance between the imaging devices and the object based on elemental images. In the reconstruction process, 3D images can be obtained by two different methods: optical reconstruction and computational reconstruction. In optical reconstruction (i.e., 3D display), a 3D image can be displayed in 3D space by display elemental images through the homogeneous lenslet array in the pickup process. However, in optical reconstruction, since the resolution of elemental images is generally low due to a lot of lenslets and the resolution limit of the display panel, the visual quality of the 3D images may not be sufficient for analyzing 3D information. Therefore, to analyze 3D information more accurately, a computational reconstruction method may be considered.

Three-dimensional point clouds or three-dimensional sliced images with different depths can be reconstructed by computational reconstruction algorithms. Volumetric Computational Reconstruction (VCR) is the most fundamental technique in the computational reconstruction process [23,25]. Figure 2 describes the principle of VCR. In optical reconstruction, optical errors (i.e., optical aberration and diffraction) may cause the degradation of the visual quality of the reconstructed 3D image. On the other hand, VCR uses virtual pinholes to enlarge and superimpose elemental images at the reconstruction plane without any optical errors. Consequently, more accurate 3D images can be reconstructed than the optical reconstruction. Also, the computational reconstruction may remove the occlusion in front of objects. In addition to VCR, computational reconstruction includes the Pixel of Element images Rearrangement Technique (PERT) [32], which accelerates reconstruction process speed, PERTS (PERT considering projected empty space) and convolutional PERTS (CPERTS), which improve the optical reproducibility of PERT [33,34]. In this paper, we consider VCR as the computational reconstruction algorithm.

In Figure 2, each elemental image is projected onto the reconstruction plane at the reconstruction depth $z_{r}$ through a virtual pinhole array, and then all elemental images are superimposed with their own shifting pixels, $S_{x},S_{y}$ which is defined as the following [23,25]:(1)$$S_{x}=\frac{N_{x}pf}{c_{x}Z_{r}},\;S_{y}=\frac{N_{y}pf}{c_{y}Z_{r}}$$
where $N_{x},N_{y}$ are the number of pixels for each elemental image in *x* and *y* directions, *p* is the pitch between pinholes, *f* is the focal length or distance between the elemental image and virtual pinhole, $c_{x},c_{y}$ are the image sensor sizes in the *x* and *y* directions, and $Z_{r}$ is the reconstruction distance. Finally, at the reconstruction depth, the 3D image $R(x,y,Z_{r})$ can be reconstructed by the following [23,25]:(2)$$R\left(x,y,Z_{r}\right)=\frac{1}{O(x,y,Z_{r})}\sum_{i=1}^{N_{x}}\sum_{j=1}^{N_{y}}E_{ij}\left\{x-S_{x}\left(i-1\right),y-S_{y}\left(j-1\right)\right\}$$
where $E_{ij}$ is the *i*th column and *j*th the row elemental image, i,j are the index of elemental images, x,y are the pixel indices of each elemental image, and $O(x,y,Z_{r})$ is the overlapping matrix at reconstruction depth. However, it may be difficult to reconstruct the accurate 3D information of objects under extremely low light conditions due to the lack of photons. Therefore, photon-counting integral imaging can be utilized to obtain 3D information of objects in these conditions.

### 2.2. Photon-Counting Integral Imaging

Human eyes and cameras can capture the image of an object by contacting the light reflected by the object. For this reason, under photon-starved conditions, the object may not be captured accurately. To solve this problem, photon-counting integral imaging was proposed [35]. It is a technique to visualize 3D images under extremely low light environment by estimating photons with statistical methods such as maximum likelihood estimation (MLE) or maximum a posterior (MAP), which is called Bayesian estimation. In this paper, we use the MLE method only. The photons are extracted statistically based on Poisson random process because photons occur rarely in unit time and space, and it means that photons are extracted from 3D object under these conditions randomly. A photon-limited image (i.e., an image recorded by photon-counting integral imaging) can be obtained by the following [35]:(3)$$\lambda \left(x,y\right)=\frac{I(x,y)}{\sum_{x=1}^{N_{x}}\sum_{y=1}^{N_{y}}I(x,y)}$$
(4)$$C\left(x,y\right)|\lambda \left(x,y\right)=Poisson[N_{p}\times \lambda \left(x,y\right)]$$
where I(x) is an image taken under normal light intensity. As a result, the normalized image λ(x,y) is obtained by Equation (Equation 3), which means that the normalized image has a unit energy. In addition, $N_{p}$ is the number of extracted photons, and C(x,y) is the photon-limited image. The number of extracted photons can be set at any value, and the brightness of the photon-limited image depends on the number of extracted photons. For a color image, photon-counting integral imaging can be implemented as follows [36]:(5)$$N_{p}=\gamma_{c}W=\frac{\eta \times W}{h\times \nu_{c}}$$
(6)$$\gamma_{c}=\frac{\eta }{h\times \nu_{c}}$$
where *W* is the energy incident on the photosensitive surface during measurement, *c* is the color channels (i.e., Red, Green, and Blue), *h* is Planck’s constant, $\nu_{c}$ is the average spatial frequency for each color channel, and η is the quantum efficiency, which is the average number of photovoltages generated by each incident photon (η≥1). In Equations (Equation 3)–(Equation 6), photon-counting integral imaging has the problem that results are random due to the feature that Poisson random numbers are temporally and spatially independent.

Figure 3 shows the problem of the photon-counting integral imaging through the result of executing photon-counting twice on the same original image. In the enlarged part of the result image, it is noticed that the pattern of the pixels is different each other. Due to this randomness, photon-counting integral imaging may not visualize a correct photon-limited image. To solve this problem, in this paper, we apply Kalman filter to photon-counting integral imaging.

### 2.3. Kalman Filter

Kalman filtering is a technique that can correct uncertain data in state-space models [27]. The Kalman filter is used for time-varying vehicle position estimation, rocket trajectory estimation, and in-car navigation. In this way, a Kalman filter is generally used to correct time-series data. As we mentioned earlier, conventional photon-counting integral imaging has the problem that the output results are random. Therefore, in this paper, photon-counting integral imaging is executed multiple times for the same original image using a Kalman filter.

Figure 4 illustrates the principle of the Kalman filter. As shown in Figure 4, the pixel values of multiple photon-limited images at the same coordinates are grouped into one array. This assembled array is regarded as the time-series data in a Kalman filter. A Kalman filter can be divided into two steps: the prediction step and the update step. In the prediction step, the time-series data of the next state is predicted from the time-series data of the current state. Equations of the prediction steps can be expressed as follows [27]:(7)$$x_{k+1}^{\alpha \beta }=x_{k}^{\alpha \beta }+v_{k},\;\{\begin{array}{c} \alpha =1,2,3,\cdots ,N_{x}\\ \beta =1,2,3,\cdots ,N_{y} \end{array}$$
(8)$$p_{k+1}=p_{k}+\omega_{k}$$
where $x_{k}^{\alpha \beta }$ is the initial pixel value, $x_{k+1}^{\alpha \beta }$ is the pixel value predicted from the initial value, and $v_{k}$ is the noise pixel value. In addition, $p_{k}$ and $p_{k+1}$ are the initial and predicted variances, and $\omega_{k}$ is the variance noise. In this paper, we use the same value for $v_{k}$ and $\omega_{k}$. This model is called a local model Kalman filter. After the prediction step, the predicted value is corrected in the update step. Equations of the update steps can be expressed as follows [27]:(9)$$G_{k}=\frac{1}{B}\sum_{e=1}^{B}\frac{p_{k}^{e}}{p_{k}^{e}+v_{k}}$$
where $G_{k}$ is the Kalman gain. In addition, *B* is the total number of photon-counting iterations. $G_{k}$ is determined as the average value of the results calculated a total of *B* times. The predicted pixel value $x_{k+1}^{\alpha \beta }$ is corrected by the Kalman gain as follows [27]:(10)$${\hat{x}}_{k+1}^{\alpha \beta }=x_{k+1}^{\alpha \beta }+G_{k+1}\times \left(y-x_{k+1}^{\alpha \beta }\right)$$
where *y* is the time-series data, which is the maximum value of the array that stores the pixel values of the same coordinates. The corrected $x_{k+1}^{\alpha \beta }$ in this step becomes the new predicted pixel value. At the end of the update step, the variance used in the calculation of Kalman gain is corrected. The equation for calculating the variance $p_{k+1}^{e+1}$ corrected from the predicted value is as follows [27]:(11)$$p_{k+1}^{e+1}=\left(1-G\right)\times p_{k+1}^{e},\;\left(e=2,3,4,\cdots ,B\right)$$

Equations (Equation 9)–(Equation 11) are repeated for all time-series and coordinates. In other words, in the next time-series, ${\hat{x}}_{k+1}$ is newly set to $x_{k+1}^{\alpha \beta }$, and $p_{k+1}^{e+1}$ is set to $p_{k+1}$, and prediction and update are repeated. The ${\hat{x}}_{k+1}^{\alpha \beta }$ for all the coordinates finally obtained in this way is the corrected photon-limited image from the execution results of multiple indeterminate photon-counting integral imaging.

Figure 5 illustrates the flow chart of the Kalman filter. The corrected photon-limited image can increase the accuracy of the photon intensity compared to the conventional elemental image and maintain photon intensity consistently. Therefore, it can be visualized even in extremely low light conditions. The image corrected by the proposed method can be reconstructed into a 3D image by Equation (Equation 2).

Figure 6 shows the comparison results of conventional photon-counting technology and the Kalman filter. For both results, the number of photons in 10% of the total number of pixels was applied. As shown in Figure 6, conventional photon-counting elemental images may not consistently and accurately visualize objects with a Poisson distribution. Accordingly, pixel intensity may decrease through the reconstruction process. On the other hand, the proposed method can accurately visualize objects through Kalman filtering, which can maintain accurate photon intensities in images. Therefore, the proposed method can more accurately visualize 3D objects even under extremely low light conditions.

## 3. Image Processing and Experimental Setup

In this paper, we propose two methods to compare the visualization results and processing time due to differences in the locations where Kalman filtering is applied in the VCR algorithm.

### 3.1. Kalman Estimation before Reconstruction (KEBR)

The first method is to apply Kalman estimation before VCR. We call this method KEBR (Kalman Estimation Before Reconstruction). Figure 7 illustrates an overview of the KEBR method. Photon-counting integral imaging is performed for each elemental image for the extremely low-light image. These photon-limited images are regarded as time-series data in Kalman filtering, and the photon-limited images arranged side by side are treated as an array. This array is corrected by Kalman filtering. By performing correction, it is possible to generate a group of photon-counting elemental images that are more accurate than conventional photon-counting images. After that, using the photon-counting elemental images corrected by Kalman filtering, reconstruction by VCR and noise removal by the median filter are performed [37].

### 3.2. Kalman Estimation after Reconstruction (KEAR)

The second method is to apply Kalman estimation after VCR. We call this method KEAR (Kalman Estimation After Reconstruction). Figure 8 describes an overview of the KEAR method. As shown in Figure 8, the KEAR method applies Kalman filtering to the process of overlapping and averaging pixels in the conventional VCR. After that, denoising was performed using the median filter on the image reconstructed by Kalman filter.

### 3.3. Experimental Setup

Figure 9 shows the experimental setup in this paper. As shown in Figure 9, two objects at different distances from the camera are located. Both objects are plastic figures, and the xyz dimensions of objects 1 and 2 are 25 mm × 55 mm × 30 mm and 30 mm × 55 mm × 40 mm, respectively. The imaging method has the same effect as a camera array and uses a device called an XYZ stage that can move the camera at equal intervals, and acquires elemental images.

Table 1 shows the specification and setup of the image sensor in this experiment. For the reconstruction distance $Z_{r}$ in photon-counting integral imaging, the distance from the camera to object 1 is 150 mm, and the distance from the camera to object 2 is 200 mm. Also, the pitch *p* of the camera moved on the XYZ stage is set to 3 mm in both the horizontal and vertical directions. In this experiment, we recorded 5×5 elemental images, for a total of 25 elemental images. For both proposed methods KEBR and KEAR, we applied the number of photons in 5 steps of 2, 4, 6, 8, and 10% of the total number of pixels, and compared the visualization results between the conventional method and the proposed method. In addition, in this experiment, *B* is 30, and $v_{k}$ and $\omega_{k}$ are 1000, respectively.

## 4. Experimental Results and Analysis

Figure 10 shows captured elemental image. Figure 10a shows an example of an elemental image obtained under normal light conditions, and Figure 10b shows an example of an elemental image obtained under extremely low light conditions. As shown in Figure 10b, it can be seen that no object can be recognized with the naked eye under extremely low light conditions.

Figure 11a,b show the reconstructed 3D images at 150 mm and 200 mm under normal light conditions, respectively. To verify the effectiveness of our proposed method, in this paper, it is compared with the conventional photon-counting integral imaging. In this paper, the method of applying the conventional photon-counting integral imaging to elemental images is called PCBR (photon-counting integral imaging before reconstruction), and the method of applying it to the reconstruction image is called PCAR (photon-counting integral imaging after reconstruction).

Figure 12 shows the experimental results. Figure 12a–d show the result images of the conventional photon-counting integral imaging and Figure 12e–h show the result images of our proposed methods. Figure 12a,b are the results of PCBR method, Figure 12c,d are the results of PCAR method, Figure 12e,f are the results of KEBR method, and Figure 12g,h are the results of KEAR method. In addition, the number of the extracted photons applied to the result images is 2% of the total number of pixels. As a result, we can recognize that the two proposed methods in this paper not only reduce random noise but also show a brighter and more accurate shape of the object. In addition, the results of the proposed methods have the effect of appearing brighter than the results of conventional photon-counting integral imaging, even though the same amount of photons is extracted. This is because the conventional photon-counting elemental image generates a lot of random noise and those pixels do not overlap in the reconstruction process, resulting in reduced brightness, and elemental image by Kalman filtering has little random noise and does not darken in the reconstruction process. When we compare the processing times, the KEBR method can be processed in about 1800 s and the KEAR method in about 60 s. This is because the KEBR method performs photon-counting imaging 25 times for each elemental image, while the KEAR method performs photon-counting integral imaging only once for each elemental image. For a more accurate comparison, we conducted numerical analysis using metrics such as structure similarity (SSIM), peak signal-to-noise ratio (PSNR) and cross-correlation [38,39,40].

Figure 13 shows the result of the numerical analysis. Figure 13a,b show the result of SSIM, and Figure 13c,d show the result of cross-correlation. In addition, Figure 13e,f show the result of PSNR. As shown in Figure 13a,b, both proposed methods in this paper show better results than both conventional photon-counting integral imaging in SSIM. On the other hand, it can be seen that the cross-correlation value exceeds the KEAR method value as the number of photons extracted by PCBR increases, as shown in Figure 13c,d,f. However, it can be said that the proposed method is more efficient because the value of the KEAR method is also high enough and the data processing time is much lower.

As a result, both proposed methods can generate high-quality 3D images even under extremely low light conditions. The proposed algorithms optimize the accurate photon intensity value through the observed photon intensity. They update the variance error and estimate photon intensity from the photon data under various light conditions. Finally, the proposed methods can generate high-quality 2D and 3D images using Kalman filtering, even under extremely low light conditions.

## 5. Conclusions

A representative limitation of conventional 3D imaging technology is that it is difficult to visualize in an extremely low light condition. Photon counting has been used to overcome these limitations, but inaccuracy of information due to random noise is a problem. In this paper, we have proposed a novel image processing method to solve this 3D visualization problem under extremely low light conditions using Kalman filtering. We have proposed two methods: the KEBR method, which applies Kalman filtering to elemental images, and the KEAR method, which applies photon-counting integral imaging only once to elemental images and applies Kalman filtering to reconstructed images. In addition, we have compared the proposed methods with conventional photon-counting integral imaging. As a result, both methods have shown better results than the conventional method. However, while the KEAR method has had a shorter processing time than the KEBR method, there have been no significant differences in the numerical comparison results. Therefore, we propose that it is better to use the KEBR method when a more accurate object shape is required, even if the processing speed is slow, and the KEAR method when a faster processing speed is required. In this paper, 3D visualization has been performed using the extremely low-light image, but there is a problem that the shadowed part of the object may not be visualized well. In the pixel values of the extremely low-light image, although there is a small amount of numerical information in the areas other than the shadows, most of the pixel values in the shadows are zero. Thus, visualization by photon-counting integral imaging may not be possible. Since the proposed method is based on photon-counting technology, it has a limitation that photons cannot be extracted from a black object, which is the limit of photon-counting technology, but there is no limitation regarding the size and type of the object. A machine-learning approach is considered to be the best way to solve this problem. It is thought that more accurate repair of hidden parts can be performed by learning data that are a set of original images and visualized images. The proposed method is expected to contribute to the development of overall technologies that require photon-counting technologies such as autonomous driving systems, security camera technology, biomedical imaging, etc.

The method proposed in this paper is an image processing method used in extremely low light conditions as we mentioned earlier, and there is no reason to use this method in sufficient light conditions. However, if the video was recorded continuously under extremely low light conditions and sufficient light conditions, it would be possible to apply a method of enabling or disabling the proposed method by analyzing the brightness of the image in real time.

## Figures and Tables

**Figure 1 sensors-23-07571-f001:**
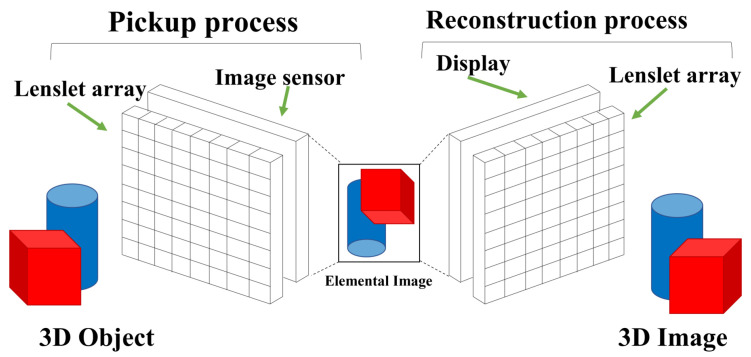
Concept of integral imaging.

**Figure 2 sensors-23-07571-f002:**
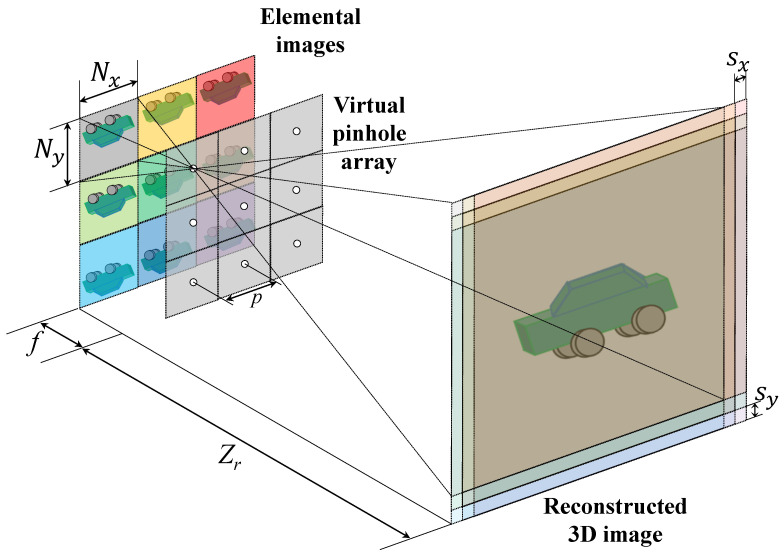
Concept of VCR.

**Figure 3 sensors-23-07571-f003:**
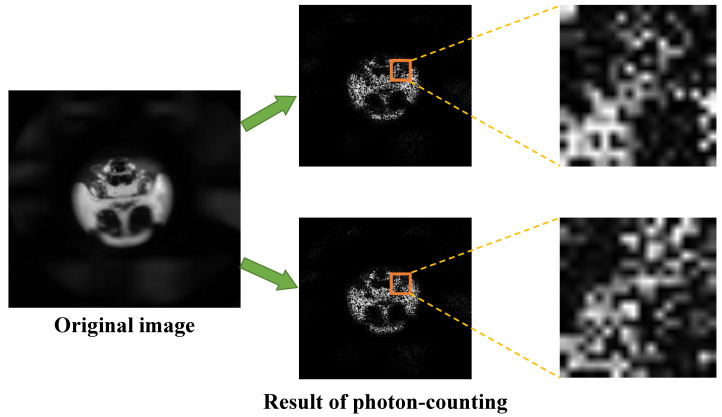
Problem of the photon-counting integral imaging.

**Figure 4 sensors-23-07571-f004:**
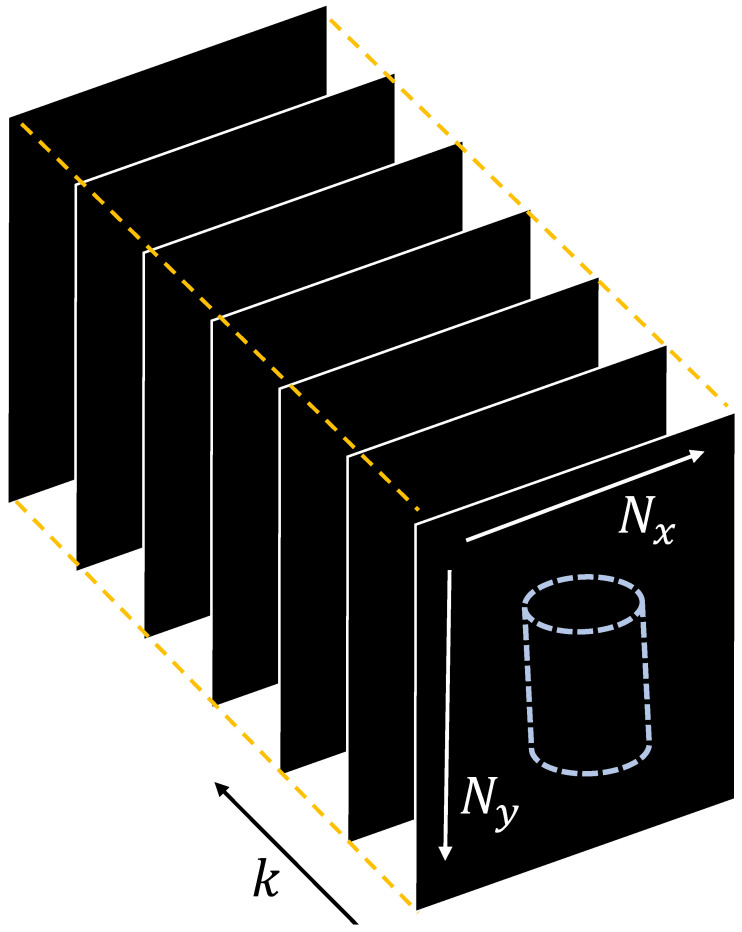
Schematic diagram of an array with multiple photon-limited images.

**Figure 5 sensors-23-07571-f005:**
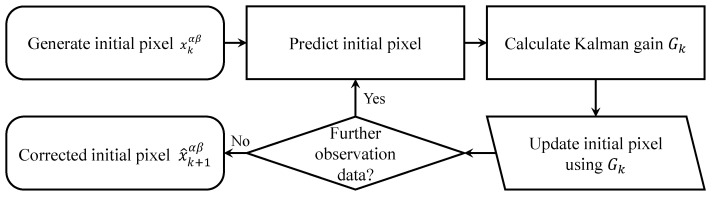
Flow chart of Kalman filter.

**Figure 6 sensors-23-07571-f006:**
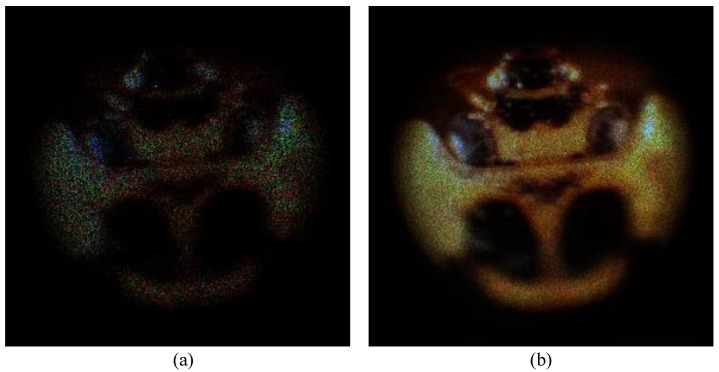
Comparison result of (**a**) conventional photon-counting technology and (**b**) Kalman filter.

**Figure 7 sensors-23-07571-f007:**
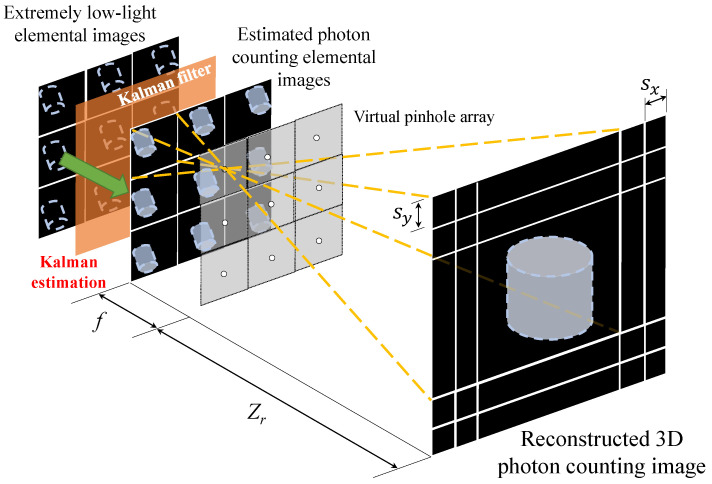
Schematic diagram of Kalman estimation before reconstruction.

**Figure 8 sensors-23-07571-f008:**
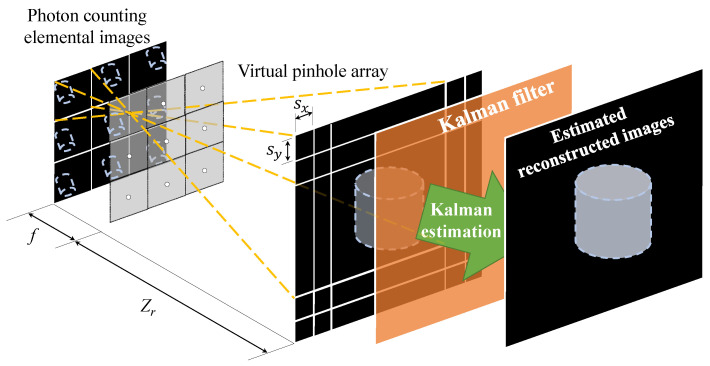
Schematic diagram of Kalman estimation after reconstruction.

**Figure 9 sensors-23-07571-f009:**
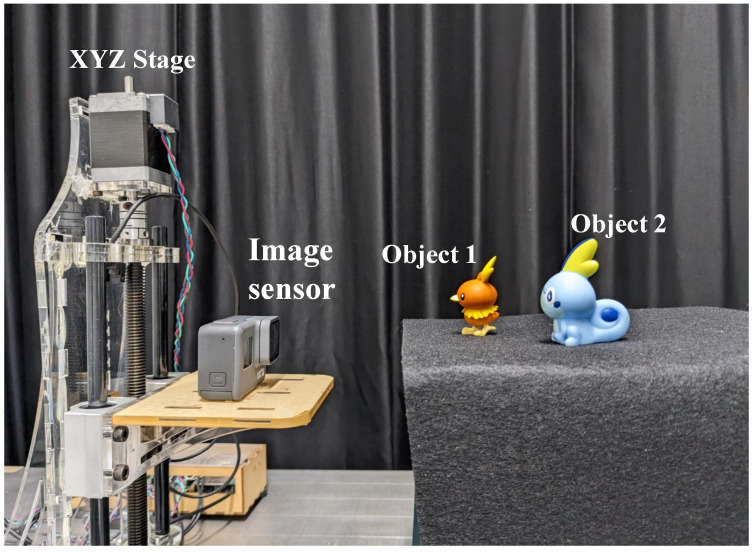
Experimental setup.

**Figure 10 sensors-23-07571-f010:**
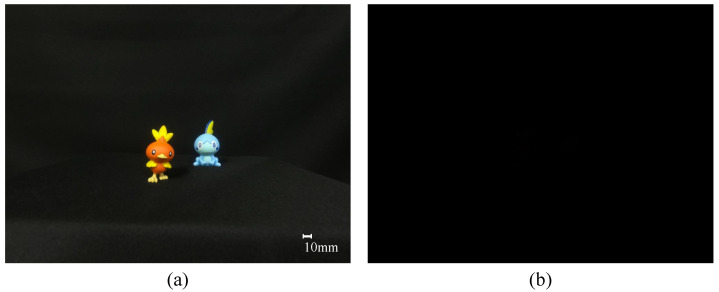
Captured elemental image of (**a**) normal and (**b**) extremely low light conditions.

**Figure 11 sensors-23-07571-f011:**
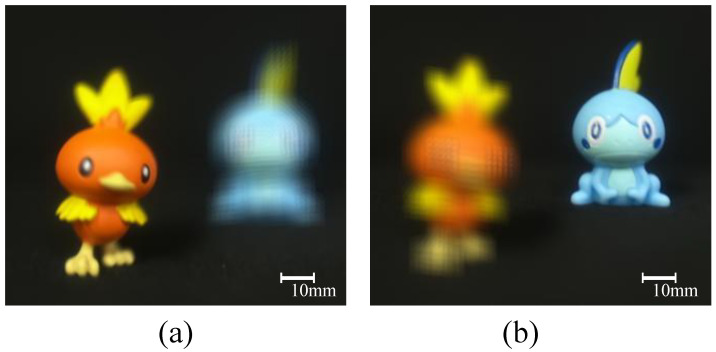
Reconstructed 3D images at (**a**) 150 mm and (**b**) 200 mm.

**Figure 12 sensors-23-07571-f012:**
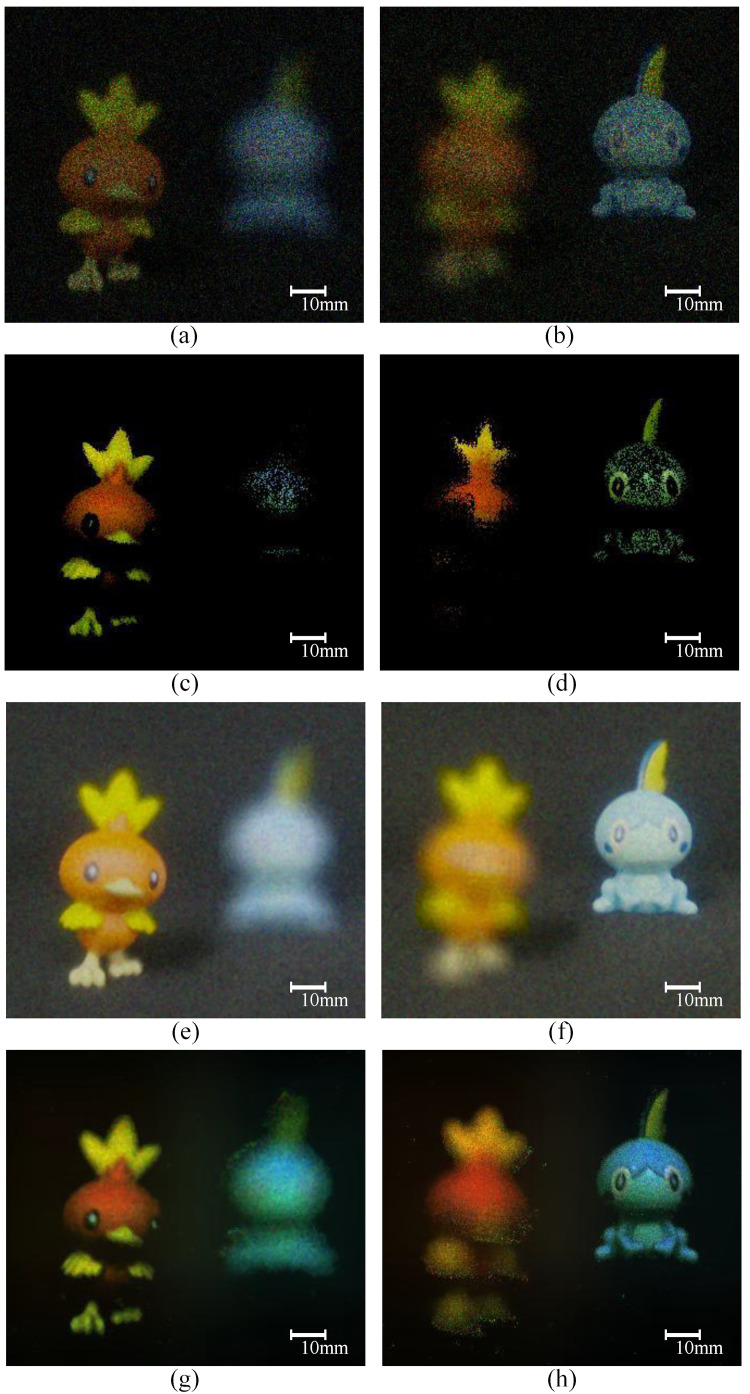
Experimental results. Images reconstructed at (**a**) 150 mm and (**b**) 200 mm after photon-counting integral imaging is applied to elemental images and images with photon-counting integral imaging applied to reconstructed images at (**c**) 150 mm and (**d**) 200 mm. In addition, Images reconstructed at (**e**) 150 mm and (**f**) 200 mm, after Kalman estimation is applied to elemental images and images with Kalman estimation, applied to reconstructed images at (**g**) 150 mm and (**h**) 200 mm.

**Figure 13 sensors-23-07571-f013:**
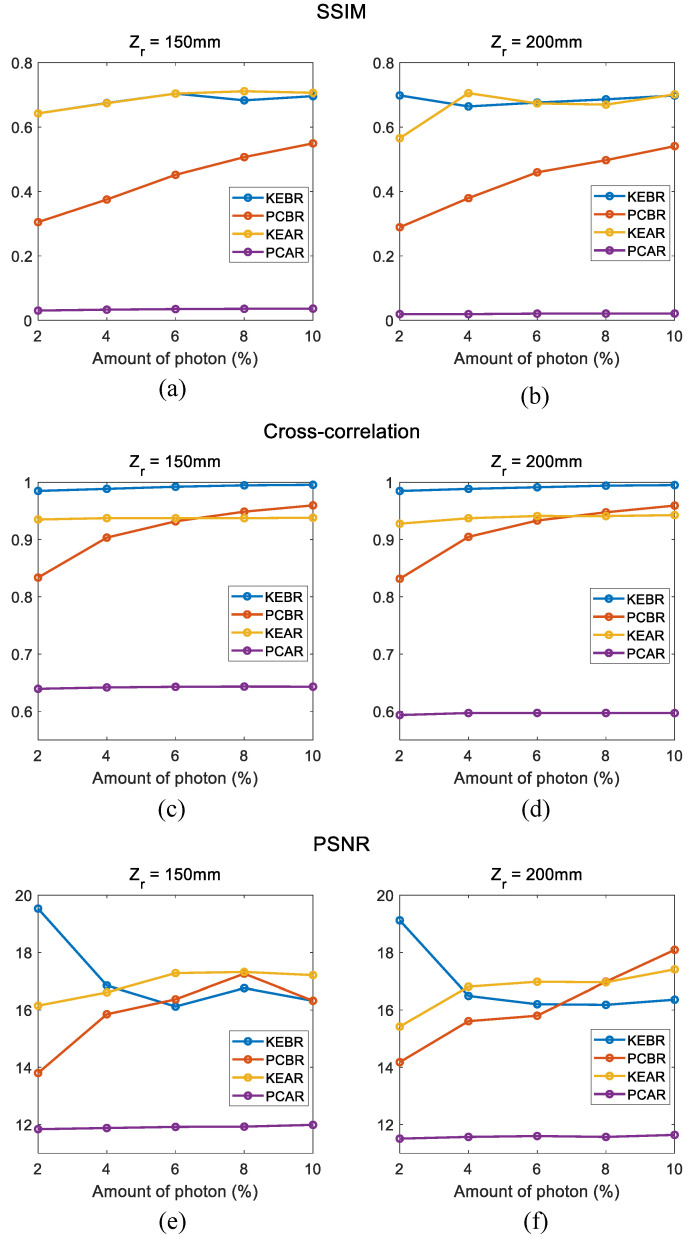
Result of the numerical analysis. SSIM of reconstructed at (**a**) 150 mm and (**b**) 200 mm, cross-correlation of reconstructed at (**c**) 150 mm and (**d**) 200 mm, PSNR of reconstructed at (**e**) 150 mm and (**f**) 200 mm.

**Table 1 sensors-23-07571-t001:** Specification of the image sensor.

Model	GoPro HERO6 Black	
Resolution ($N_{x}$ × $N_{y}$)	2000 × 1500	
Focal length (*f*)	3 mm	
Sensor size ($c_{x}$ × $c_{y}$)	5.9 mm × 4.4 mm	
Shutter speed	Normal	1/125 s
	Extremely low-light	1/2000 s
ISO	Normal	1600
	Extremely low-light	200

## Data Availability

Not applicable.

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
