# Peer review of "Three-Dimensional (3D) Visualization under Extremely Low Light Conditions Using Kalman Filter"

_sensors, 2023, doi:10.3390/s23177571_

Round 1

Reviewer 1 Report

See attachment!

Reviewer 2 Report

The authors developed a Kalman filter method to improve the visual quality of three-dimensional reconstruction image under low illumination environment. The most valuable part of this research work is that this method can solve the randomness of the photon-counting integral imaging in 3D image reconstruction. I recommend to publish this paper. However, I have some comments as below.

1. In Abstract part, the first appearance of the abbreviation PSNR and SSIM should indicate the full name.

2. Please give one more measuring example to compare the effect of 3D graphic reconstruction using Kalman filter method and the photon-counting integral imaging technology without Kalman filter method.

3. In the case of sufficient light, can the photon-counting integral imaging using Kalman filter method also have a clearer effect of 3D reconstructing the image?

Reviewer 3 Report

This work is very important and interesting. Therefore, I recommend it publication after a minor revision.

1.       Please add scale bars to Figures.

2.       Can authors add more advantages on Kalman filter?

3.       It is better to add a comparison with related works.

4.       Can authors add more descriptions on “extremely low light conditions”?

5.       To enrich the background, the authors need to add more discussions on imaging. The following references maybe useful, such as PhotoniX 3, 6 (2022); PhotoniX 4, 16 (2023); PhotoniX 3, 1 (2022); Advanced Functional Materials 33, 2208677 (2023); Nature Communications 13, 5634 (2022);
